# Dual Representation of the Auditory Space

**DOI:** 10.3390/brainsci14060535

**Published:** 2024-05-24

**Authors:** Stephanie Clarke, Sandra Da Costa, Sonia Crottaz-Herbette

**Affiliations:** Neuropsychology and Neurorehabilitation Service, Centre Hospitalier Universitaire Vaudois (CHUV), University of Lausanne, Av. Pierre-Decker 5, 1011 Lausanne, Switzerland; sandra_elisabete@hotmail.com (S.D.C.); sonia.crottaz-herbette@chuv.ch (S.C.-H.)

**Keywords:** sound localization, auditory spatial cues, primary auditory cortex, auditory belt areas, lateralization, 7T fMRI, auditory evoked potentials

## Abstract

Auditory spatial cues contribute to two distinct functions, of which one leads to explicit localization of sound sources and the other provides a location-linked representation of sound objects. Behavioral and imaging studies demonstrated right-hemispheric dominance for explicit sound localization. An early clinical case study documented the dissociation between the explicit sound localizations, which was heavily impaired, and fully preserved use of spatial cues for sound object segregation. The latter involves location-linked encoding of sound objects. We review here evidence pertaining to brain regions involved in location-linked representation of sound objects. Auditory evoked potential (AEP) and functional magnetic resonance imaging (fMRI) studies investigated this aspect by comparing encoding of individual sound objects, which changed their locations or remained stationary. Systematic search identified 1 AEP and 12 fMRI studies. Together with studies of anatomical correlates of impaired of spatial-cue-based sound object segregation after focal brain lesions, the present evidence indicates that the location-linked representation of sound objects involves strongly the left hemisphere and to a lesser degree the right hemisphere. Location-linked encoding of sound objects is present in several early-stage auditory areas and in the specialized temporal voice area. In these regions, emotional valence benefits from location-linked encoding as well.

## 1. Introduction

Behavioral and imaging studies demonstrated partial separation between neural networks underlying sound recognition and sound localization [1,2,3,4,5,6,7,8,9,10,11,12,13,14,15,16,17,18]. The dichotomy between the two networks was confirmed by the conscious experience of patients with focal lesions. Patients with predominantly temporal lesions tended to present deficits in sound recognition, whereas those with predominantly parietal lesions presented deficits in sound localization [19,20,21,22,23,24]. The latter involved often the right hemisphere [25,26,27,28], documenting right hemispheric dominance in respect to the ability to localize sounds in space. A series of studies highlights, however, a more complex concept of the auditory space, with a right-lateralized representation supporting explicit use and a strongly left-lateralized representation contributing to implicit use. The present review summarizes the arguments for the latter.

Our investigation into the field started with a clinical observation. A young patient was consulted 9 years after she suffered a large right hemispheric stroke that caused massive left hemispatial neglect and deprived her of the ability to localize sounds [29]. She described vividly her difficulties in everyday life. When crossing the street, she needed to check visually several times where the car she heard was and how fast it approached—she simply could not rely on her spatial hearing. When listening to a conversation in a group, she was uncertain who was speaking, and she could not identify where the speaker was. She scanned the group and looked for lips moving or hoped to recognize the person who spoke by her/his voice. In contrast to these major difficulties was the fact that the patient worked as cashier in a supermarket, without apparent discomfort. She did not experience any problems understanding what customers were telling her, despite the very loud surroundings. Our assessment of auditory cognitive functions revealed severe deficits in sound localization and in detection of sound motion. For both aspects, the spatial dimension was simulated by interaural time differences (ITD); the patient was not able at all to discriminate azimuthal positions, indicating all sound sources arbitrarily at a central position. Neither could she differentiate between moving and stationary sounds. Despite these major deficits, the patient was able to use auditory spatial cues to segregate simultaneous sound sources. To test this aspect, ITD cues were used to implement a spatial-release-from-masking paradigm. The target sound was the cry of a tawny owl, always presented at a central position (ITD = 0 µs). The masker sound was a helicopter sound presented at one of eleven possible spatial positions simulated with ITD (400, 320, 240, 160, 80 µs to the left or right, or 0 µs). Normal subjects do not perceive the target sound when the masker is at the same, central position. By contrast, they perceive the target when there is spatial distance between the target and the masker. For this test, the patient performed as a normal subject, which indicated that she was able to use spatial cues to separate the target from the masker. This clinical observation suggested that there are at least two partially independent networks that encode auditory spatial cues: one for the explicit ability to localize consciously the sound sources, and the other for covert use in discrimination of sound sources on the basis of their location.

We review here evidence pertaining to brain regions, which are involved in the implicit representation of the auditory space. Unlike sound localization, which requires explicit indication of the perceived location of the sound source by word or deed, implicit use of auditory spatial cues contributes to sound object segregation and to the tracking of individual sound objects across space, even in the absence of conscious sound source localization. It relies on the combined encoding of sound meaning and location. The latter has been investigated in neuroimaging studies and referred to as location-linked representation of sound objects.

## 2. Methods

### 2.1. Search Strategy

The primary search strategy concerned functional magnetic resonance imaging (fMRI) studies. To this effect, we carried out a systematic search of the electronic database PUBMED (Advanced Search Results—PubMed (nih.gov) with the term ((((((“spatial” [Title/Abstract] OR “location” [Title/Abstract] OR “position” [Title/Abstract]) AND “auditory” [Title/Abstract] AND “fMRI” [Title/Abstract]) NOT “speech” [Title/Abstract]) NOT “prosody” [Title/Abstract]) NOT “language” [Title/Abstract]) NOT “schizophrenia” [Title/Abstract]) NOT “psychiatric” [Title/Abstract] on 6 February 2024 and a last check on 15 March 2024. This search was performed twice This strategy yielded 457 citations, which were checked on the basis of their title and abstract and, when relevant, the full text for inclusion and exclusion criteria. Twelve original papers were identified in this way.

An additional search for relevant papers using magnetoencephalography (MEG) or auditory evoked potentials (AEP) was carried out in the electronic database PUBMED with the same search strategy, but with “MEG” or “AEP” instead of “fMRI”. The search yielded 211 citations for MEG and 62 for AEP, which were checked on the basis of their title and abstract and, when relevant, the full text for inclusion and exclusion criteria; one AEP study was thus identified.

### 2.2. Inclusion Criteria

Studies involving healthy adult subjects without history of brain damage, psychiatric disorders, and/or hearing impairment and using meaningful auditory stimuli were included when they compared activations elicited by (i) the same sound object in changing vs. stationary location or (ii) location and identity changes in a 2-factorial design. MEG and AEP studies were considered when they provided whole brain source estimations.

### 2.3. Data Analysis

Twelve fMRI studies included lists of MNI or Talairach coordinates. Activation clusters were visualized by means of BrainNet Viewer (http://www.nitrc.org/projects/bnv/, accessed on 20 March 2024 [30]) several times during February 2024.

## 3. Results

### 3.1. Focal Brain Lesions: Dissociation between Explicit and Implicit Use of Auditory Spatial Cues

The initial clinical observation [29] motivated further investigations into impairments of explicit sound localization and implicit use of spatial cues. Duffour-Nikolov and colleagues assessed 13 patients during the post-acute or early chronic stages of a first hemispheric lesion, caused by stroke or traumatic brain injury, for their ability to use auditory spatial cues explicitly in sound localization and implicitly for spatial release from masking [31]. Both aspects were investigated with ITD as spatial cues. The authors reported double dissociation between explicit vs. implicit use of spatial cues, i.e., six cases with impaired explicit and preserved implicit use, as well as one case with preserved explicit and impaired implicit use. They also found different types of sound localization deficits associated with preserved implicit use. These results emphasize the independence of neural representations supporting sound localization and location-linked representation of sound objects.

Tissieres and colleagues investigated the anatomical correlates of deficits in sound localization and of deficient performance in spatial release from masking [32]. A total of 40 patients during the post-acute or chronic stages of a first unilateral focal hemispheric lesion were included, 20 with left and 20 with right lesions. Deficits in implicit sound localization correlated with lesions of the right parietal and opercular cortex, as reported previously [27]. Spatial release from masking was investigated with a paradigm involving the detection of a centrally presented target (here: owl cry). The subjects failed to detect the target when the masker (helicopter sound) was presented at the same, central location but did so successfully when the masker was presented at a lateral location (while the target was presented always at the central location). The inability to benefit from spatial separation between the target and the masker was correlated with lesions of the left temporo-parieto-frontal cortex or of the right inferior parietal lobule and underlying white matter. These results document the partial anatomical segregation of neural networks underlying explicit and implicit use of auditory spatial cues. Furthermore, they emphasize the role of the left hemisphere in location-linked representation of sound objects.

Supporting evidence for a left hemispheric contribution to the implicit representation of auditory space is provided by anatomical correlates of auditory extinction. Tissieres and colleagues investigated anatomical correlates of dichotic (two different words, one in each ear) and diotic (two different words, lateralized with interaural time differences to the right and to the left) listening in a series of 39 patients during the post-acute or early chronic stages of a focal unilateral stroke of the left or the right hemisphere [33]. Unilateral extinction, i.e., the failure to report words presented on one side in the di-condition (but not in the mono-condition), occurs after contralateral lesions, mostly in the context of unilateral neglect. Bilateral extinction is pathognomonic of the failure to use auditory spatial cues for sound object segregation. It was correlated with lesions within a large parieto-fronto-temporal region of the left hemisphere or with lesions within a smaller parieto-temporal region of the right hemisphere. These results document the contribution of the left hemisphere to the implicit use of auditory spatial cues for sound object segregation.

### 3.2. Neuroimaging Paradigms to Investigate Location-Linked Representation of Sound Objects

The ability to use spatial cues to segregate sound objects is believed to rely on neural representations, which combine sound object identity and its location [34,35]. Behavioral studies have shown that spatial cues facilitate speech perception [36,37], even in situations where subjects do not attend to specific locations or targets [38,39,40,41,42,43,44]. The effect of spatial cues is believed to enhance the early processes of stream segregation. Eramudugolla and colleagues have shown that the discrimination of animal cries and musical instruments among distractors benefits from spatial cues when the target sound is short (250 ms) but not long (500 ms) [45]. In contrast to the above quoted studies, which used meaningful sounds as targets, the contribution of spatial cues to the segregation of meaningless, previously unheard sound events is less clear [46,47].

Investigating the location-linked representation of sound objects requires the use of environmental sounds and comparison between conditions in which a given sound object changes its location vs. remains stationary. Several studies compared the effect of changing location to a stationary condition (paradigm I in Figure 1) [48,49,50,51], whereas other studies investigated the effect of location vs. identity changes in a 2-factorial design (II in Figure 1) [52,53,54,55,56,57]. We review here the evidence from these studies in the context of location-linked representation of sound objects. It is to be noted that further studies used meaningless sounds with similar paradigms [11,18,58,59,60,61,62,63,64,65] or focused on the effects of binaural interactions [66,67,68,69], but are not directly relevant to the issue discussed here.

### 3.3. AEP: Left Hemispheric Contribution to the Rapid Discrimination of Location Changes

The specificity of neural representations has often been investigated with adaptation paradigms. A neural population, which encodes a specific feature of a stimulus, decreases its activity when this feature remains constant across repeated presentations, but increases it again when this feature changes [70,71,72]. Repetition suppression in response to a repeated presentation of sound objects has been documented by electrophysiological recordings as a decrease in evoked potential amplitude and shown to occur within a critical time window post-stimulus onset [73,74].

Using the repetition paradigm with auditory evoked potentials (AEP), Bourquin and colleagues explored how the spatial attributes of individual sound objects are encoded [49] (Figure 2). Individual environmental sounds were presented twice in a continuous series. The initial presentation was lateralized by means of interaural intensity differences either to the right or left hemifield; the repeated presentation followed after zero to five intervening sounds and was lateralized to the same or to the opposite side of the initial presentation. AEP analysis revealed differences between repetition effects when the sound location was shifted vs. held constant between the initial and repeated presentations. These effects were significant at 20–39 ms post-stimulus onset within a cluster on the posterior portion of the left inferior and middle temporal gyri, and at 143–162 ms on the left inferior and middle frontal gyri, providing evidence for a location-linked representation of sound objects. Thus, neural populations within two left-hemispheric regions detect rapidly the change in location of a specific sound object. 

### 3.4. fMRI Evidence for Combined Encoding of Sound Identity and Location

A series of fMRI studies investigated auditory representations by combining environmental sounds and their spatial attributes and reported contrasts relevant to location-linked representation of sound objects (Table 1).

#### 3.4.1. Changing vs. Fixed Locations

Five studies compared activation elicited when sound objects changed location vs. remained stationary (Figure 2, with four studies using a whole-brain approach [48,51,54,75] and one focusing on the supratemporal plane [53]. Two further studies investigated the interaction between the emotional valence of sounds and their location [55,56]. It is to be noted that goals differed between studies. Two investigated neural correlates of keeping track of sound objects in space [53,55], and another one investigated correlates of sound object segregation [51]. Four studies focused on differences between sound identity and location differences [54,73], or emotional valence and location differences [56], or on spatial attention [75]. Because of this variety, we shall summarize for each study the intended goal, the methodology, and results pertaining to the contrast of changing vs. fixed locations.

Brunetti and colleagues set out to investigate the auditory *Where* pathway with a passive listening paradigm. They presented the sound of a knife as it taps on a glass at five azimuthal positions across the right and left space [50]. In a given run of eight presentations, the five locations were mixed or remained fixed at either the left- or the right-most location. The comparison of activations elicited by stimulus presentation at mixed locations as compared to presentations at either the left or right position yielded foci on the posterior part of the superior temporal gyrus in either hemisphere (Table 1; Figure 3A). In accordance with the aim of the study, only one sound object was used throughout the experiment. Thus, it remains uncertain whether the effect reflected a location-linked representation of sound objects or the first stages of explicit sound localization.

Altmann and colleagues set out to investigate the effects of selective attention to sound identity or to sound location [48]. They used cries from 10 animal species at two locations on the left and two locations on the right; pairs of sounds combining the same or different object identity and location were presented and the subjects were asked to match sound identity or location. The contrast of different vs. same location was calculated as activation elicited by the [condition with different location and identity + condition with different location and same identity]—[condition with same location and different identity—condition with same location and identify] and was used in further analysis. It highlighted foci on the planum temporale and the posterior part of the superior temporal sulcus in either hemisphere, as well as in the right anterior insula (Table 1; Figure 3A). These results provide evidence for the neural underpinning of spatial changes of meaningful sounds, without, however, addressing the issue of tracking specific sound objects in their changes of location.

Smith and colleagues set out to investigate the impact of the number of sound objects on the encoding within the planum temporale [51]. They presented one or three talkers in three spatial conditions: (i) same location, (ii) changing between locations, or (iii) moving smoothly from one side to the other. The contrast of one talker alternating between three locations vs. one talker at one location was calculated to define ROIs for further analysis. This specific contrast yielded foci in the left and right planum temporale (Table 1; Figure 3A), which are indicative of location-linked representation of sound objects.

Grady and colleagues investigated age differences in neural adaptation to sound identity and location and used human non-speech, animal, musical and machine sounds at five locations [54]. A given trial consisted of four stimuli, all of which belonged to one of the four possible configurations of same/different sound identity and same/different location. In the young control population, the contrast of “same sound at different locations” > “different sounds at the same location” revealed several foci predominantly in the fronto-temporal cortex, with only a relatively small bilateral contribution of the supramarginal gyrus in either hemisphere (Table 1; Figure 3A). There is a striking paucity of parietal involvement, considering that the contrast involved different > same locations.

Da Costa and colleagues studied how early-stage auditory areas integrate sound object meaning and location and focused on the supratemporal plane, which is described in detail in the following section [53].

In summary, the combined representation of sound identity and location, as explored by activation elicited by location changes of meaningful stimuli, involves specific early-stage auditory areas [53] as well as fronto-temporal regions [50,51,54,73] bilaterally. There is a striking absence of right parietal involvement (Figure 3A).

#### 3.4.2. Other Location Paradigms

Four other studies used environmental sounds at changing locations with experimental paradigms, which did not involve a direct comparison with the same sound object when stationary (Figure 3B; Table 1). Bidet-Caulet and colleagues investigated neural correlates of listening to a walking human [75]. The comparison of active localization of steps vs. noise detection revealed foci within the superior temporal sulcus on either side. Brunetti and colleagues compared the sound of a knife tapping on a glass presented at different locations with rest, which yielded significant clusters bilaterally on Heschl’s gyrus and on the right side in the posterior part of the superior temporal gyrus, inferior parietal lobule and prefrontal cortex [76]. Altmann and colleagues investigated the processing of location and pattern changes, using two types of animal cries presented on the right or on the left [52]. Experimental blocks involved sound object and/or location changes. The main effect of location changes highlighted bilateral clusters in the superior temporal lobe. Zündorf and colleagues set out to investigate correlates of sound localization in complex acoustic scenes [77]. Active localization of a target in a cocktail party-type auditory scene, as compared to the passive listening of the same scene, yielded significant clusters bilaterally on the superior temporal gyrus as well as in the left supplementary motor area, anterior insula and inferior frontal gyrus, and in the right frontal eye field.

Taken together, these four studies, which used diverse location paradigms, highlight bi-hemispheric involvement in the combined encoding of sound identity and location. Surprisingly, they also reveal paucity of parietal involvement. Both these aspects emphasize the difference between the neural correlates of the sound identity and the neural underpinning of auditory localization, which involves predominantly the right parietal cortex [27].

#### 3.4.3. Contribution of Early-Stage Auditory Areas

Da Costa and colleagues used an adaptation paradigm with 7T fMRI in a 3-factorial design, which was implemented in blocks of 2 × 4 sounds [53]. First, sounds were either eight different exemplars of the same sound object or eight different sound objects (factor Category). Second, they were lateralized to the right or left hemifield by means of interaural time differences (factor Location). Third, there were two types of blocks, with or without location changes between the first and the second set of four sounds (factor Location change). The analysis was carried out with 3-way ANOVA Category × Location × Location change and focused on the supratemporal plane. This design aimed at detecting neural populations, which encode the combined representation of sound meaning and of its location; these populations are expected to decrease their activity during the second part of the block if the sound object remains at the same location, and increase it when the sound object changes position (Figure 4A). Da Costa and colleagues investigated neural activity in individual auditory areas, which were identified by means of tonotopic mapping [78] and comparison with locations described in two prior anatomical studies [79,80]. Two non-primary auditory areas on the left hemisphere, and one primary and one non-primary area on the right hemisphere, yielded a significant interaction Category × Location change, driven by greater activity after Location change (Figure 4B). These results document the presence of neural populations encoding location-linked representation of sound objects at the level of early-stage auditory areas.

Three other studies reported significant effects on the supratemporal plane. Using sounds of different animal species at two left and two right locations, Altmann and colleagues compared activations elicited by the four combinations of same/different species and same/different location [48]. Comparing the two conditions with location change (i.e., with same or different species) to those without location change (again for both the same or different species), they identified significant clusters bilaterally on the planum temporale. The comparison with the published coordinates of individual auditory areas [53,55,79] indicates that both clusters involve the postero-lateral area L1 (Figure 4B).

Brunetti and colleagues used a single sound object, a knife tapping on a glass, presented either at a mix of five azimuthal locations or at the left-most or right-most location [75]. When comparing the activation elicited by the mix of location with the rest condition, they identified four significant clusters. Two were on the right hemisphere, within the R and A1 subdivisions of the primary auditory cortex, and the other two were on the left hemisphere, within the postero-medial area M1 and the lateral area L2 (Figure 4B). 

Smith and colleagues compared activation elicited by one talker at three locations with one talker at one location [51]. This contrast yielded one significant cluster on the right and one on the left planum temporale, situated between the posterior areas M1 and L1 (Figure 4B). 

Taken together, these four studies [51,53,73,75] document the presence of neural populations encoding location-linked representation of sound objects at very early stages of cortical processing.

### 3.5. fMRI Evidence for Location-Linked Encoding of Emotional Valence

Evidence from fMRI studies indicates that location impacts the encoding of emotional valence. Kryklywy and colleagues investigated the effect of emotion during auditory localization in a two-factorial paradigm with emotional sounds of positive, neutral and negative valence, presented at four different locations [56]. Two-way ANOVA Location × Emotion revealed significant interaction in three regions: (i) on the right supratemporal plane, driven by greater activity elicited by positive and negative as compared to neutral stimuli when presented on the left, and (ii) in the bilateral precuneus and medial occipital lobe, driven by greater impact of location on positive and negative but not neutral stimuli (Figure 3). Comparison with previously published coordinates of early-stage auditory areas [53,81] indicates that the right supratemporal cluster is approximately within the lateral area L2 (Figure 5). These results speak in favor of combined encoding of emotional valence and spatial information. In a second study, Kryklywy and colleagues re-analyzed their data with the multivoxel pattern analysis searchlight approach [57]. Neural activity elicited by localization of emotional sounds was searched for activity patterns predictive of sound location and/or emotion. Areas predictive of location were identified on the supratemporal plane, in the general regions of the primary auditory cortex (Figure 5).

Grisendi and colleagues used an adaptation paradigm with 7T fMRI with two sound categories (human vocalization, other environmental sounds), three valences (positive, neutral, negative) and three locations (left, center, right) [55]. Significant interaction between the factors valence and location was present bilaterally in the voice area. Thus, the spatial origin of sound modulates the encoding of emotional valence.

Taken together, these findings speak in favor of location-linked encoding of emotional valence. Currently, we can only speculate about behavioral significance and propose two lines of interpretation.

First, spatial cues may render emotional stimuli more salient. In a previous study, Grisendi and colleagues used the same set of stimuli in a 7T fMRI study, which involved a similar experimental paradigm [81] with two sound categories (human vocalization, other environmental sounds) and three valences (positive, neutral, negative), but no spatial cues. The same analysis in terms of tonotopically defined auditory areas was carried out as in Grisendi et al., 2023 [55]. The comparison of significant effects of valence in either study showed that the mere presence of spatial cues enhanced the differential processing of emotional valence within early-stage auditory areas. When no spatial cues were used, the main effect of valence was present in two areas on the right and two on the left anterolateral temporal plane [81]. When stimuli were presented with spatial cues, six additional areas on either side yielded the main effect of valence (Figure 4) [55]. These results offer an explanation for the previously reported enhanced intensity and arousal associated with emotional sounds presented in space [82,83,84,85].

Second, the combined representation of emotional valence and location may confer distinct emotional values to specific parts of space. Thus, Grisendi and colleagues have shown that human vocalizations with positive valence, which were presented on the left side, yielded greater activity in the ipsilateral and contralateral primary auditory cortex than did neutral or negative vocalizations or any other stimuli at any of the three locations (Figure 5) [55]. Location-linked encoding of emotional valence is very likely the neural underpinning of the previously reported left-ear advantage for identifying emotion [86,87,88].

## 4. Discussion

### 4.1. Current Evidence for Location-Linked Representation of Sound Objects

The implicit representation of the auditory space, the counterpart of the explicit sound localization, relies on the location-linked encoding of sound objects. Several lines of evidence demonstrate the independence of the two representations of the auditory space. Double dissociations observed in investigations with brain-damaged patients indicate that the implicit and explicit use of spatial cues depends on largely distinct neural networks [31]. Whereas explicit sound localization relies predominantly on the right hemisphere [27], implicit use involves strongly the left and to a lesser degree the right hemisphere [32,33].

The combined representation of the identity and the location of sound objects involves very early stages of cortical encoding within the left hemisphere [49], which is likely to enhance the early processes of stream segregation and lead to better discrimination in the presence of distractors [45]. Beyond the early stages of cortical processing, the combined representation of sound identity and location is supported bilaterally by early-stage auditory areas and by fronto-temporal regions [48,50,51,52,53,54]. 

Location-linked representation of sound objects is likely to contribute to memory representations. Attending to memory traces, when the sound object is no longer present, was proposed to involve two distinct pathways, one guided by attention to higher order features and the other by attention to sensory information [89]. As demonstrated in behavioral studies, both the explicit and implicit long-term memory for the location of a target modulates subsequently deployed auditory spatial attention [90]. Evidence from AEP investigations indicates that implicit encoding of auditory spatial cues is at play [91]. 

Location also impacts the encoding of emotional valence, with effects in early-stage auditory areas as well as in temporal and parietal regions [55,56,57]. Spatial cues appear to make emotional stimuli more salient [55,81]. Furthermore, the combined representation of emotional valence and location may confer distinct emotional values to specific parts of space [55].

### 4.2. Outstanding Issues

Future studies need to investigate several aspects of the implicit representation of the auditory space. First, we do not currently know which cortical regions are critical for the implicit use of auditory spatial cues and at what time point of processing. This aspect has been investigated for explicit sound localization by means of chronometric single-pulse transcranial magnetic stimulation [92]. For the implicit representation of the auditory space, this approach would help to understand the relative contributions of the left vs. right hemisphere. Second, current methodological approaches give us insight into the intra-areal organization of human auditory areas, either with histological techniques in post-mortem human tissue or with ultra-high field fMRI in vivo. Both approaches reported columnar and laminar characteristics. Layers III and IV of the primary auditory cortex were shown to comprise 500 µm wide stripes, characterized by a high vs. low level of cytochrome oxidase activity and oriented perpendicularly to isofrequency lines [93]. Intrinsic connections, as traced in post-mortem tissue with the carbocyanine dye DiI, involve narrow parts of the cortex within the primary auditory areas, but spread over larger parts in belt areas [94]. 7T fMRI studies revealed differences in auditory encoding across cortical layers, including increasing complexity of encoding in superficial layers [95,96], stability of frequency tuning across cortical columns but not across cortical layers [97], and different depth profiles of auditory vs. visual inputs [98]. It is currently unknown which cortical layers contribute to the combined encoding of sound identity and location, and how early in cortical processing meaning of a sound object and its location are linked. Third, we do not know whether the implicit auditory representations are dynamic and can be altered by training, as can spatial representations supporting explicit sound localization [99]. Further studies would also need to clarify if explicit and implicit use of auditory spatial cues can substitute for each other in cases of selective deficits.

### 4.3. Clinical Impact

The presence of a dual representation of the auditory space should be taken into account in patient populations who experienced normal hearing before suffering hearing impairment later in life. They are likely to have a dual representation of the auditory space and would benefit greatly from hearing aids, which provide combined information about the meaning and location of individual sound objects. Such an approach may be of interest to patients with autosomal dominant non-syndromic hearing loss, whose hearing impairment often starts after at least one decade of normal hearing [100].

### 4.4. Virtual Reality and Artificial Intelligence Applications

Virtual reality (VR) and artificial intelligence (AI) applications are likely to benefit from location-linked representation of sound objects to provide a more captivating and real-world experience for subjects. This approach could be used for AI application in otology [101]. The principle of a dual representation of a variable, here space, may have an impact beyond neural encoding of the auditory space. Diagnostic procedures by means of machine learning are likely to benefit from AI models to predict risks of developing a severe medical condition [102]; they may be implemented in future specific links between variables. 

## 5. Conclusions

Implicit representation of the auditory space contributes to location-linked representation of sound objects and thus plays a major role in sound object segregation and in attention. Evidence from lesion and neuroimaging studies demonstrated strong left-hemispheric involvement with a weaker contribution from the right hemisphere. Location-linked encoding of sound objects occurs during the first stages of cortical processing and involves early-stage auditory areas. It is noteworthy that not only do concrete sound objects benefit from location-linked encoding, but also emotional valence.

Future studies should address the fine organization of the implicit representation of the auditory space and determine how malleable it is. Since location-linked representation of sound objects is a key feature of auditory processing, clinical applications such as hearing aids should specifically combine semantic and spatial information about distinct sound sources. Similarly, combined semantic and spatial information about specific sound objects is likely to benefit virtual reality and artificial intelligence applications.

## Figures and Tables

**Figure 1 brainsci-14-00535-f001:**
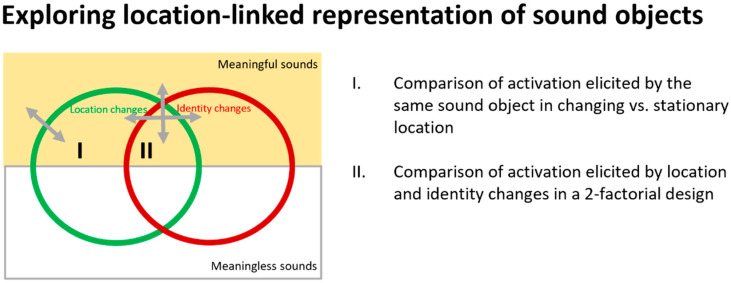
Experimental approaches to location-linked representation of sound objects. Schematic representation of relevant stimulus characteristics.

**Figure 2 brainsci-14-00535-f002:**
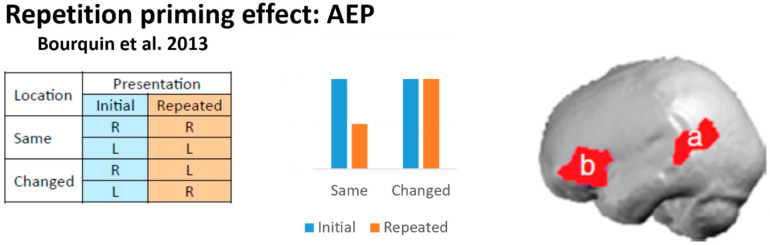
Repetition priming effect in electrophysiological studies. **Left**: Design of the study comparing AEP elicited by the initial and the repeated presentation of a sound object remaining at the same location or changing location. **Middle**: Expected AEP magnitude for the initial and repeated presentation at the same or changed location. **Right**: Using distributed source modeling of AEP, Bourquin and colleagues [49] identified two regions with significant effects, at 20–39 ms post-stimulus onset at the posterior superior and middle temporal gyri and at 143–162 ms on the left inferior and middle frontal gyri (marked a and b on brain figurine). L: left; R: right.

**Figure 3 brainsci-14-00535-f003:**
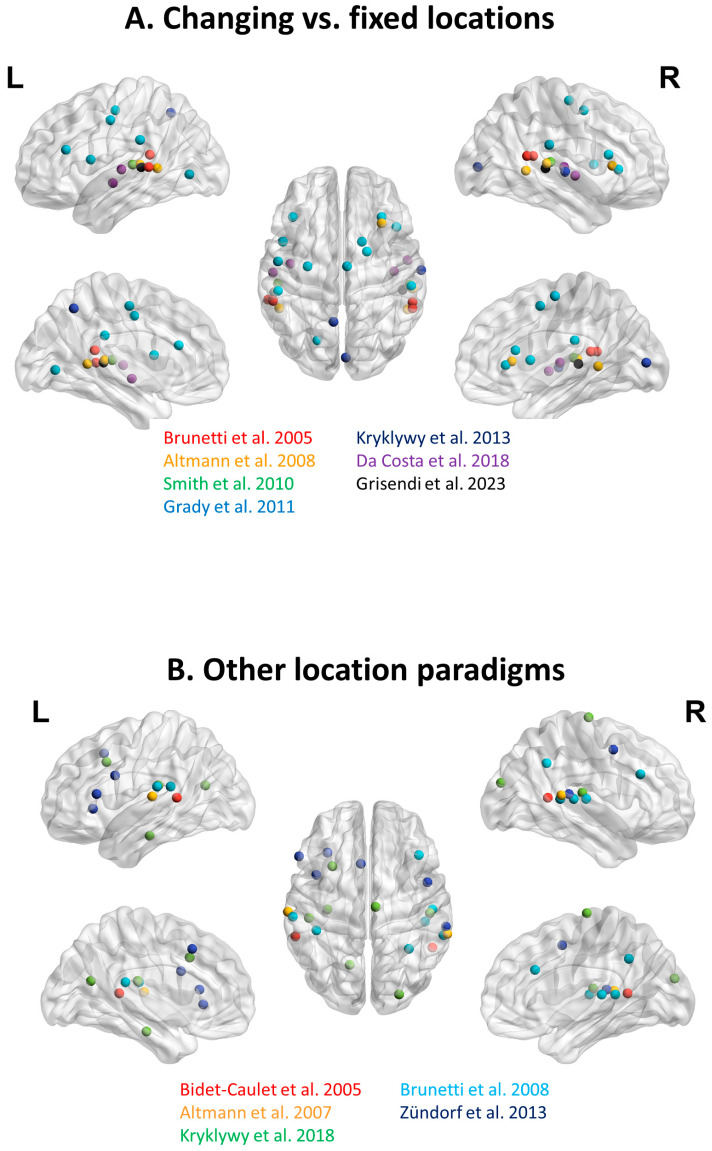
Neuroimaging studies investigating the representation of environmental sounds by comparing changing vs. fixed location [48,50,51,53,54,55,56] (**A**) or in other location-relevant paradigms [52,57,75,76,77] (**B**). Activation clusters reported in individual studies with indications of coordinates (Table 1) were visualized by means of BrainNet Viewer (http://www.nitrc.org/projects/bnv/, accessed on 20 March 2024) [30]. For details of studies, see Table 1.

**Figure 4 brainsci-14-00535-f004:**
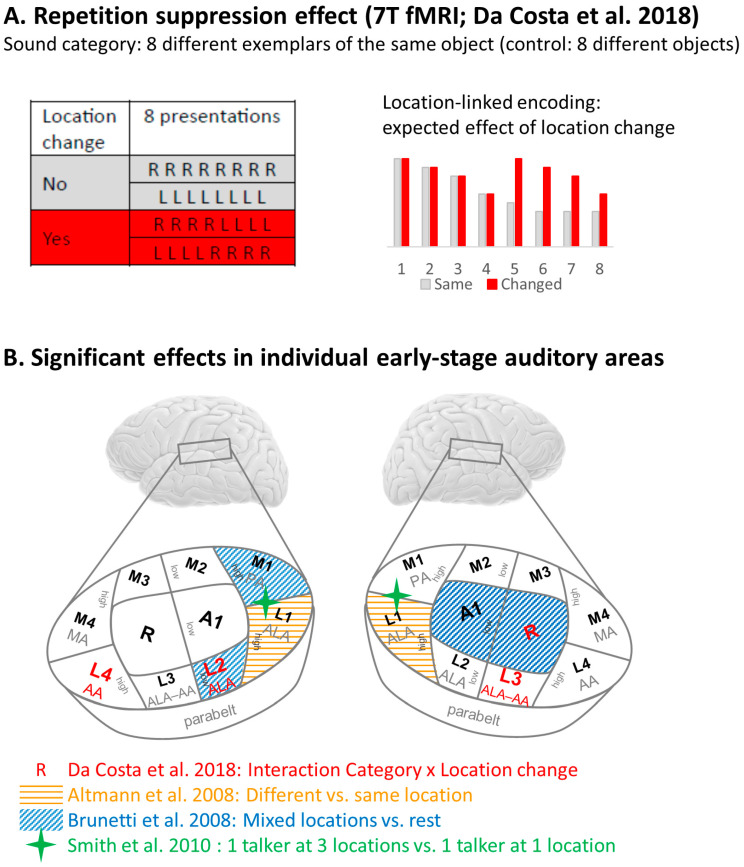
Location-linked representation of sound objects within early-stage auditory areas. (**A**) Repetition suppression effect used in our 7T fMRI study [53]. Three-factorial design was used, with factors Category, Location and Location change (for details see text). If present, the location change occurred between the fourth and the fifth presentation of a sequence of eight (highlighted in red in left panel). Expected response by neuronal population which keeps track of the location of a given sound object (highlighted in red in right panel). (**B**) Schematic representation of early-stage auditory areas summarizing significant effects reported in individual studies, including primary auditory cortex (A1, R), medial belt areas (M1, M2, M3, M4), lateral belt areas (L1, L2, L3, L4) and the parabelt region [48,51,53,75]. Low and high regions of frequency gradients (as determined by tonotopic mapping [78]) and cytoarchitectonically defined areas (AA, ALA, MA, PA; [79,80]) are indicated. For more details of individual studies, see text and Table 1.

**Figure 5 brainsci-14-00535-f005:**
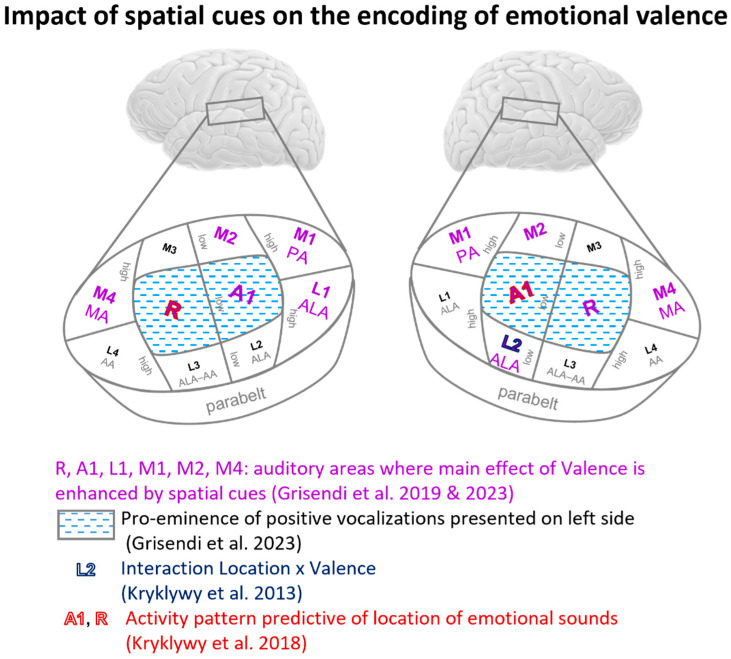
Impact of spatial cues on the encoding of emotional valence [55,56,57,81]. Schematic representation of early-stage auditory areas, including primary auditory cortex (A1, R), medial belt areas (M1, M2, M3, M4), lateral belt areas (L1, L2, L3, L4) and the parabelt region. Low and high regions of frequency gradients (as determined by tonotopic mapping, [78]) and cytoarchitectonically defined areas (AA, ALA, MA, PA; [79,80]) are indicated. For more details of individual studies, see text and Table 1.

**Table 1 brainsci-14-00535-t001:** Spatial representation of meaningful sound objects. Summary of significant effects demonstrated in activations studies (fMRI) using paradigms with active localization, passive listening to changes in location, or adaptation to location. Methodology and key conclusions as reported by authors. Coordinates are indicated in MNI space (transformed if in Talairach in the original publication). Brodmann’s areas (BA), brain regions (br), or auditory areas (aa) as indicated in original publications. L = left, R = right. Brain region descriptions: a = anterior; FEF = frontal eye field; HG = Heschl’s gyrus; INS = insula; p = posterior; PFC = prefrontal cortex; put = putamen; TG = superior temporal gyrus; STL = superior temporal lobe; STS = superior temporal sulcus. Auditory areas as in Da Costa et al., 2018 [53]; VA = voice area.

Publication	Methodology	Contrast & Conclusions	Side	BA/br/aa	x	y	z
Bidet-Caulet et al., 2005 [75] Listening to a walking human activates the temporal biological motion area Neuroimage	3T fMRI 10 normal subjects Footsteps - One walker on left - Another walker right to left Task: indicate direction of crossing walker	**Active localization:** Footstep task > noise detection **Conclusions:** Posterior STS activation by human steps in line with its role in social perception	R	STS	52	−50	9
			L	STS	−50	−41	9
Brunetti et al., 2005 [50] Human Brain activation during passive listening to sounds from different locations: an fMRI and MEG study Human Brain Mapping	1.5T fMRI 11 normal subjects Knife tapping on glass −90°, −45°, 0°, 45°, 90° Runs 8 presentations: - Mixed (5 locations) - Left (−90°) - Right (90°) Task: passive listening	Mixed vs. right	R	p STG	57	−41	15
			L	p STG	−56	−40	16
		Mixed vs. left **Conclusions:** Both hemispheres involved in processing sounds from different locations	R	p STG	57	−46	15
			L	p STG	−60	−39	6
Altmann et al., 2007 [52] Processing of location and pattern changes of natural sounds in the human auditory cortex Neuroimage	3T fMRI—event-related 17 normal subjects Sheep, dog −90°, 90° Blocks of 9 presentations with sound object and/or location changes Task: passive listening	Location changes **Conclusions:** Posterior STL involved in spatial processing of auditory stimuli	L	STL	−71	−21	10
			R	STL	64	−39	11
Altmann et al., 2008 [48] Effects of feature-selective attention on auditory pattern and location processing Neuroimage	3T fMRI adaptation 12 N subject 10 animal species −70°, −15°, 15°, 70° Pairs of sounds Task: match sound identity or location	Different vs. same location (different location and identity plus different location minus different identity minus same location and identity) **Conclusions:** fMRI adaptation effects in STS, PT and INS for location changes	L	p STS	−51	−46	5
			R	p STS	53	−47	3
			L	PT	−53	−32	8
			R	PT	57	−30	9
			R	a INS	32	23	7
Brunetti et al., 2008 [76] A frontoparietal network for spatial attention reorienting in the auditory domain: a human fMRI/MEG study of functional temporal dynamics Cerebral Cortex	1.5T fMRI 10 normal subjects Knife tapping on glass Alternate locations: Right: 90°, 50° Left: −90°, −50°, Central: −20°, 20° Mixed: −90°, −50° 0°, 50°, 90° Task: passive listening	Mixed vs. rest **Conclusions:** Supratemporal plane modulated by variations in sound location	R	HG	52	−19	8
			L	HG	−61	−25	18
			R	HG	46	−29	8
			L	HG	−42	−36	18
			R	STG	61	−40	8
			R	IPL	33	−51	37
			R	PFC	41	25	28
Smith et al., 2010 [51] Auditory spatial and object processing in the human planum temporale: no evidence for selectivity J. Cognitive Neuroscience	3T fMRI 10 normal subjects 1 or 3 talkers 1 location; changing between 3 locations; moving −60° to 60° Task: passive listening	Spatial manipulation (3-locations/1-talker vs. 1-location/1-talker) **Conclusions:** Spatial sensitivity in PT reflects auditory source separation using spatial cues	L	PT	−52	−26	8
			R	PT	55	−27	10
Grady et al., 2011 [54] Age differences in fMRI adaptation for sound identity and location Frontiers Neuroscience	3T fMRI 19 young/20 old normal subjects Sounds (S): human non-speech, animal, musical, machine Location (L): −95°, −60°, 0°, 60°, 95° Conditions: - Same S, same L - Same S, different L - Different S, same L - Different S, different L Task: to detect repetition	Adaptation to location (in young): Same sound, different locations > different sounds, same location **Conclusions:** Age differences in adaptation to repetition of sound location	R	6	20	0	52
			R	45	44	20	16
			R	47	32	28	4
			L	44	−48	8	12
			L	46	−40	28	20
				6	4	−12	60
			L	6	−28	−12	52
			L	6	−52	−8	44
			R	40	56	−28	24
			L	40	−52	−32	28
			L	18	−20	−72	0
			R	put	16	8	8
Kryklywy et al., 2013 [56] Emotion modulates activity in the what but not where auditory processing pathway Neuroimage	3T fMRI 18 normal subjects Emotional sounds in space: 4 locations × 3 valences Task: to localize sounds	2 ANOVA location X valence Significant interaction **Conclusions:** Emotion modulates activity in the «what» but not the «where» auditory processing pathway	R/L	31, 7	−7	−57	50
			R/L	19, 18, 17	3	−86	6
			R	42, 41, 22	64	−15	2
Zündorf et al., 2013 [77] Neural correlates of sound localization in complex acoustic environments PLoS ONE	3T fMRI 20 normal subjects Environmental sounds Location: −45°, −22.5°, 0°, 22.5°, 45° Conditions: Single: 1 target at one of 5 locations Cocktail: 1 target among 5 different sounds at different locations Passive: as cocktail, no task Sequence: 1–5 sounds consecutively, no location Task: to localize the target	Cocktail > Passive **Conclusions:** Activity related to auditory stream segregation in posterior STG, INS, SMA and fronto-parietal network	R	STG	63	−33	12
			L	STG	−66	24	12
			L	SMA	−6	18	45
			L	a INS	−33	27	0
			L	IFG	−42	9	27
			R	FEF	48	3	48
Da Costa et al., 2018 [53] Keeping track of sound objects in space: the contribution of early-stage auditory areas Hearing Research	7T fMRI 10 normal subjects Block of 2 × 4 sounds: Same or different category Locations: Left (−60°, −40°) or right (40°, 60°) Location change: yes or no Task: passive listening	3-way ANOVA Category × Change in location × Location Interaction Category × Change in location Driven by larger effect for change in “same” category and for no-change in “different” category (mean coordinates of tonotopically identified auditory areas in individual subjects) **Conclusions:** A third auditory stream, originating in lateral belt areas, tracks sound objects across space	L	L2	−56	−17	4
			L	L4	−42	−10	−7
			R	R	42	−16	6
			R	L3	53	−7	−1
Kryklywy et al., 2018 [57] Decoding auditory spatial and emotional information encoding using multivariate versus univariate techniques Experimental Brain Research	3T fMRI 18 normal subjects 12 sounds with neutral, positive or negative valence Location: −90°, −22.5°, 22.5°, 90° Task: to localize sounds with specific emotion	Multivariate search light analysis Activity patterns predictive of sound location **Conclusions:** Multivariate pattern analysis larger overlapping spatial and emotional representation of sound within early secondary auditory regions than univariate analysis	L	2, 4, 13, 41, 42	−48	−26	19
			R	2, 4, 13, 41, 42	50	−22	13
			L	8	−29	16	38
			L	31	−15	−64	19
			R	24	6	−17	47
			R	19	25	−88	21
			L	35	−33	−19	−22
Grisendi et al., 2023 [55] Emotional sounds in space: asymmetrical representation within early-stage auditory areas Frontiers Neuroscience	7T fMRI 13 normal subjects Category: human vocalizations; environmental sounds Emotional valence: positive, neutral negative Location: −60°, 0°, 60°) Task: passive listening	3-way ANOVA Category × Valence × Location Significant interaction Valence × Location (mean coordinates of auditory areas identified by tonotopic mapping or voice area localizer in individual subjects) **Conclusions:** Positive vocalizations presented on left side yield strong activity; spatial cues render emotional valence more salient within early-stage auditory areas	L	VA	−55.50	−33.45	6.08
			R	VA	48.79	−31.39	5.46

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
