# Peer review of "Dual Representation of the Auditory Space"

_brainsci, 2024, doi:10.3390/brainsci14060535_

Round 1

Reviewer 1 Report

Comments and Suggestions for Authors

The authors reviewed the concept of auditory space lateralisation based on the explicit and implicit use. It was an interesting discussion and may help those who want to have a quick summary of the topic.

Comments:

1. For a review article, a proper methodology on the literature search need to be described as well

2. In the abstract, the aims of the review and the methods need to be mentioned.

3. No need citations in the conclusion

4. For future studies, it is better to bring into the discussion part as it does involve a lot discussion and citations

Author Response

We thank the Reviewer for her/his insightful comments, which we found very useful. We have revised the manuscript accordingly and the different points were addressed as described in the enclosed document (the Reviewer’s comments are in italics, our responses in blue; the relevant changes in the manuscript are highlighted in blue).

Reviewer 2 Report

Comments and Suggestions for Authors

Abstract:

- The abstract provides a summary of the paper's major ideas, although it is unclear in other places and leaves some important facts. Here are some ideas for improvement:

- In the opening phrase, clearly define the goal or purpose of the review paper. At the moment, it summarizes findings without giving any background.

- Describe the differences between the "implicit" and "explicit" uses of auditory spatial cues. Give definitions to the terms.

Steer clear of imprecise terms such as "seminal" and instead describe the nature of prior research with greater specificity.

- While neuroimaging investigations are mentioned in the abstract, the MRI and EEG methodologies employed are not specified. It is necessary to have this context.

- Make it clear if the results speak of hemispheric dominance or differences. This is perplexing right now.

Expand upon the patient case study that served as the inspiration for this review. List the main deficiencies that were discovered.

Saying "recent studies" is not as descriptive as mentioning the quantity and kind of studies reviewed.

Introduction

- The opening gives background information but is unfocused. It would gain from

- Giving the reader an early orientation by providing a succinct synopsis of the subject and objectives of the review.

- More information about pertinent earlier research, properly cited. Steer clear of ambiguous terms like "seminal studies". discuss and cite doi:10.3390/biomedicines11061616

- Giving more patient information;

- Extending the clinical case that served as the basis for this review.

- At the conclusion of the introduction section, clearly define the purpose and parameters of the current review.

Methods:

- The methods section has to be added; it is currently lacking. It ought to contain:

- Information on the search terms, databases used, and inclusion/exclusion criteria for the literature.
- The quantity and kinds of research that are part of the review (e.g., x patient studies, y neuroimaging, etc.).

- Techniques for synthesizing and analyzing data from the examined studies.

The quality of the review would be greatly enhanced by the addition of a methodology section.

Results:

- The outcomes are not properly organized or structured. Advice:

- Rather of bouncing between study kinds, divide into parts according to study type (neuroimaging, lesion studies, etc.).

Don't merely describe studies one by one; instead, provide a more structured summary of the major conclusions from each study inside each section.

Increase the number of headings and subheadings in the content to help the reader navigate it.

- For a brief summary, include a table that summarizes the key conclusions and methodology of each neuroimaging study.

Steer clear of giving each study's methodology in the results text unnecessary detail.

Discussion

- The debate lacks depth and several important details that should be in a review paper. Suggestions:

- Open with a paragraph that summarizes the main conclusions and the body of knowledge gained from this review.

- Talk about the research' shortcomings, such as methodological flaws, tiny sample sizes, a lack of replication, etc.

- Talk about how the findings affect our knowledge of auditory spatial representations and processing. Talk about artificial intelligence contribution. Discuss and cite doi:10.3390/life13030702 and doi:10.1177/0194599820931804.

- Make explicit recommendations for future study directions in light of the gaps found.

- Based on this literature analysis, conclusions should summarize what is known and what is still unclear.

Comments on the Quality of English Language

amy

Author Response

(The authors gave the same response as above.)
